# Effects of health and social care spending constraints on mortality in England: a time trend analysis

Johnathan Watkins,[1,2] Wahyu Wulaningsih,[2,3] Charlie Da Zhou,[4]
Dominic C Marshall,[5] Guia D C Sylianteng,[2,6] Phyllis G Dela Rosa,[2,7]
Viveka A Miguel,[2,8] Rosalind Raine,[9] Lawrence P King,[10] Mahiben Maruthappu[9]

JW and WW contributed equally.

For numbered affiliations see end of article.

**Correspondence to**
Dr Mahiben Maruthappu;
m.maruthappu@gmail.com and
Dr Johnathan Watkins;
jwatkins@pilar.org.uk

## ABSTRACT

**Objective** Since 2010, England has experienced relative constraints in public expenditure on healthcare (PEH) and social care (PES). We sought to determine whether these constraints have affected mortality rates.

**Methods** We collected data on health and social care resources and finances for England from 2001 to 2014. Time trend analyses were conducted to compare the actual mortality rates in 2011–2014 with the counterfactual rates expected based on trends before spending constraints. Fixed-effects regression analyses were conducted using annual data on PES and PEH with mortality as the outcome, with further adjustments for macroeconomic factors and resources. Analyses were stratified by age group, place of death and lower-tier local authority (n=325). Mortality rates to 2020 were projected based on recent trends.

**Results** Spending constraints between 2010 and 2014 were associated with an estimated 45 368 (95% CI 34 530 to 56 206) higher than expected number of deaths compared with pre-2010 trends. Deaths in those aged ≥60 and in care homes accounted for the majority. PES was more strongly linked with care home and home mortality than PEH, with each £10 per capita decline in real PES associated with an increase of 5.10 (3.65–6.54) (p<0.001) care home deaths per 100 000. These associations persisted in lag analyses and after adjustment for macroeconomic factors. Furthermore, we found that changes in real PES per capita may be linked to mortality mostly via changes in nurse numbers. Projections to 2020 based on 2009-2014 trend was cumulatively linked to an estimated 152 141 (95% CI 134 597 and 169 685) additional deaths.

**Conclusions** Spending constraints, especially PES, are associated with a substantial mortality gap. We suggest that spending should be targeted on improving care delivered in care homes and at home; and maintaining or increasing nurse numbers.

## INTRODUCTION

Health systems in most industrialised countries are facing the concurrent challenge of managing rising demand amidst funding constraints, following the recent global economic crisis. The National Health Service (NHS) in England, which provides tax-funded, universal health coverage, is no exception.[1–4] Since 2010, the NHS in England has seen a real-term annual increase in public healthcare spending of 1.30% between 2010 and 2014, as compared with historical annual growth of around 4%.[4] During the same period, demand and healthcare cost inflation have increased, with a growing and ageing population, in addition to new treatments and technologies. By 2020/2021, a funding gap, between what is needed and what is available, has been predicted, unless major changes are implemented.[4]

Although the role of social determinants in health is increasingly acknowledged,[5] there is underinvestment in social care in many high-income countries such as the USA.[6] In England, public sector funding for social care has suffered.[7] Such funds enable the provision of means-tested home care and care home accommodation,[8] allowing, for example, hospitals to discharge frail patients who would otherwise have no adequate support. Real-term adult social care spending decreased by 1.19% annually between 2010 and 2014 after correcting for the effect of inflation, reversing the annual increase of 3.17% between 2001 and 2009. This is despite increasing demand, with the group most

likely to require social care—the over 85s—set to rise from 1.6 million in 2015 to 1.8 million in 2020.[9]

**Strengths and limitations of this study**

► Time-trend analysis with projections of future outcomes and counterfactual analysis.
► Sensitivity analyses by using other outcomes: potential years of life lost and life expectancy.
► Analysis included potential effect modifiers and mediating factors.
► Variations may exist at the local level which were not identified by the study.
► There may be other mediating factors beyond those explored in this analysis.

This supply–demand mismatch has manifested in several ways. During the first week of 2017, more than 4 in 10 NHS hospitals declared a major alert.[10] Emergency medicine departments (A&E) saw 900 000 (4.6%) more attendances in 2015/2016 compared with the previous year, and 4% more emergency hospital admissions.[11] Over the past 2 years, the number of elderly patients waiting over 12 hours in A&E has trebled, and there has been a 31% increase in delayed hospital discharges.[11]

While the funding gaps facing health and social care have been well quantified,[12 13] the impact on population outcomes remains unclear. Here we sought to model the past, present and future impact of funding constraints experienced by the publicly financed health and social care system in England on mortality, to provide insights into the association between funding and health outcomes and inform future financial allocations.

## METHODS
### Data collection
#### Population mortality, potential years of life lost and life expectancy
Annual population mortality data for England were extracted from the UK's Office for National Statistics (ONS)[14] based on Medical Certificates of Cause of Death from the Registration Online system. Age-standardised death rates (ASDR) were calculated with reference to a standard European population using mortality data split into 10 age groups (see online supplementary appendix). Information on deaths by place of occurrence (care homes, hospice, home, hospital and other establishments) was provided by Public Health England. Additionally, mortality data were collected for 325 lower tier local authorities, which are the equal of districts, boroughs or city councils, based on 2010 boundaries with Cornwall and the Isles of Scilly combined, due to the small population of the latter. Data on life expectancy were obtained from the ONS, whereas data on potential years of life lost (PYLL), a measure of premature mortality from causes considered amenable to healthcare, were obtained from the UK Health and Social Care Information Centre[15] and age standardised.

#### Spending and resources data
Nominal public expenditure on health (PEH) data were defined as total expenditure limits for the Department of Health (responsible for the NHS in England), for the financial years 2001/2002 to 2014/2015, and were collated from Her Majesty's Treasury's Public Expenditure Statistical Analyses.[16] To account for inflation, real PEH using 2014/2015 financial year prices was calculated as the product of nominal PEH and the gross domestic product (GDP) deflator for a given financial year, divided by 100, which denoted the GDP deflator for the financial year 2014/2015. To ease interpretation, real PEH per capita was converted to units of £10. Public expenditure on social care

(PES) in nominal terms was defined as total gross adult social care expenditure, which excludes the Supporting People fund (a block grant to local government supporting vulnerable people living in their own homes available from 2003/2004) and NHS transfers.[17] Further details, including health and social care resources (staff and bed numbers), are provided in the online supplementary appendix.

### Statistical analysis
For all analyses, we used 2001 as the start of the study period since complete population mortality data were available from this year onwards. To assess mortality gap, time trend analyses were conducted using Poisson or quasi-Poisson regression models with all-cause ASDR as the dependent variable and calendar year as the independent variable. Analyses were stratified by sex and repeated using rates of PYLL and life expectancy as alternative health outcomes. Analysis for mortality outcomes was further stratified by age groups and place of death as well as their combination, and by local government area. Further details are provided in the online supplementary appendix.

Fixed-effects regression models were conducted using real PEH/PES per capita and controls for economic variation (unemployment[18] and the average annual consumer price index[19]) and average weekly pensions as the independent variables, and care home, home and hospital mortality as the dependent variables. For a given year, population mortality on the same year was the outcome whereas spending data for a financial year starting at the given year was used as the main predictor. Since population mortality was collected annually and spending data is reported for each financial year (starting 1 April in the UK), we repeated our analysis using 1 and 2-year lag periods. We additionally incorporated both PEH and PES in the same model.

Effects of public spending on resources such as staff or infrastructure have been documented[20] and these resources have also been linked with health outcomes.[21] Therefore, we explored resources of health and social care as potential mediating factors by running fixed-effects regression models with real PEH/PES per capita and each resource variable as the independent variables; and care home/home mortality as the dependent variable. Resource variables that were associated with the dependent variable, resulted in weaker associations between PEH/PES and the dependent variable, and retained the same coefficient sign as univariable regressions were considered to be putative mediators. Analyses were repeated for 1 and 2-year lag periods.

To investigate the possibility of a future population mortality gap, two different mortality projection analyses for 2015–2020 were performed. Each analysis had a different observation base: one using 2001–2010 data and the other using 2009–2014 data, with the number of projected deaths based on 2001–2010 data subtracted from the number of projected deaths based on 2009–2014 data.

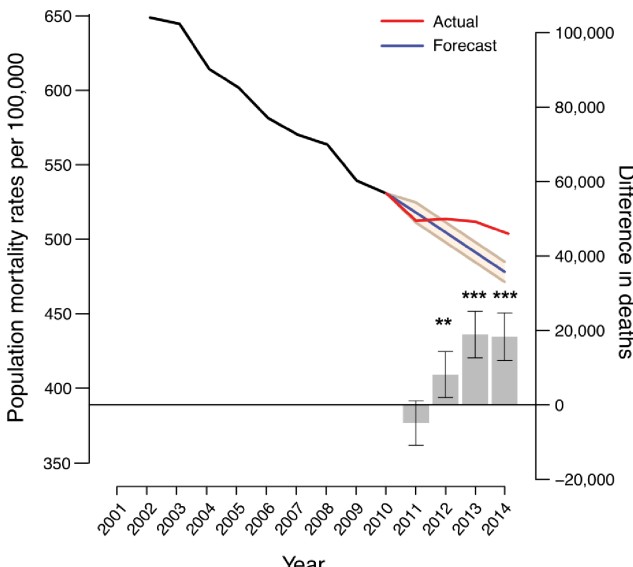

**Figure 1** Time trend projections of age-standardised death rate (ASDR) per 100 000 individuals. ASDR (left hand y-axis) and the difference in the number of deaths between actual and predicted mortality (right hand y-axis) per year from 2001 to 2014 are shown. The black and blue lines represent actual ASDR for the 2001–2010 and 2011–2014 periods, respectively. The red line represents predicted ASDR using 2001–2010 as an observation base while the 95% CIs are denoted by the beige-coloured area. The grey bars denote the differences between the number of deaths observed and the number predicted for 2011–2014 where positive values correspond to excess deaths and negative values represent lower than expected deaths. Error bars represent 95% CIs. *p<0.05; **p<0.01;***p<0.001.

Finally, using fixed-effects regression modelling, the total health and social care spending needed to close the mortality gap on top of planned spending budgets, as of 2016, was computed for three different efficiency hypothetical scenarios: 0%, 1% and 3% (see online supplementary appendix for a definition and explanation).[22]

## RESULTS
### Health and social care spending
From 2001/2002 to 2009/2010, the average annual increase in real PEH per capita was 3.82%. Between 2010/2011 and 2014/2015, the average annual increase was 0.41% (online supplementary figure S1). Planned real PEH spending to 2020/2021 is forecasted to increase at an average of 0.72% per year, based on published information up to 2016. Real PES per capita experienced an average annual increase of 2.20% between 2001/2002 and 2009/2010, while between 2010/2011 and 2014/2015 it decreased by 1.57% annually.

### Assessment of a mortality gap
From 2001 to 2010, the absolute number of deaths in England decreased by an average of 0.77% per year. From 2011 to 2014, the number of deaths increased by an average of 0.87% per year. To quantify the potential mortality

gap associated with the 2010 spending constraints, we compared actual and predicted mortality rates for both sexes and found 8148 higher than expected number of deaths in 2012 (95% CI 2004 to 14 292), 18 896 (95% CI 12 641 to 25 152) in 2013, and 18 324 (95% CI 11 953 to 24 695) in 2014 (figure 1). Sex-stratified time series analyses revealed similar results between men and women (online supplementary table S1).

To validate these results, we looked at two alternative health outcomes: life expectancy and PYLL. We found 2012–2014 male and female life expectancy to be 3.84 months (2.40–5.28) and 5.16 months (3.36–7.08) less, respectively, than the values anticipated from 2001–2003 to 2009–2011 data (online supplementary table S2). Moreover, we found that PYLL increased after the 2009/2010 spending constraints compared with predicted rates (online supplementary table S3).

To see whether the possible effects of the 2010 spending constraints were sensitive to age, we conducted time series analyses for 10 different age groups (online supplementary table S4). Higher than expected numbers of deaths were confined to those ≥60 years of age with all six ≥60 age groups showing excess deaths in 2013. By contrast, <60 age groups most often exhibited fewer deaths.

We next looked at place of death for all age groups, while stratifying them by age into <60 and those ≥60 years. Time trend analyses revealed care home and home deaths to be the first and second largest contributors, respectively, to excess deaths across age groups and in those ≥60 (figure 2 and online supplementary figure S2 and supplementary table S5), with a relative increase in these places of death over others during the study period. For every year analysed, lower than expected numbers of deaths occurred in hospitals. For those <60 years, hospitals were the most frequent place of death, and so the net result was a lower than expected number of <60 deaths. Time series analyses by local government area (for time points up to 2013) revealed no correlation between changes in the number of expected deaths and deprivation (p=0.14, $R_s$=0.08).

### Longitudinal associations between spending and mortality
We found that £10 per capita declines in real PEH and real PES were associated with increases of 0.19 (0.01–0.38) (p=0.049) and 5.10 (3.65–6.54) (p<0.001) care home deaths per 100 000 in England, respectively (table 1). For PES, these relationships persisted at least 2 years after the initial change but became weaker for PEH. We next sought to demonstrate that these associations were independent of macroeconomic forces such as unemployment (for younger age groups) and pensions (for older age groups), which have each been linked with increased mortality.[23 24] On adjusting the regression models for macroeconomic factors and the annual average of weekly pensions, we found real PES per capita and real PEH per capita to remain inversely associated with care home mortality (table 1). Next, we

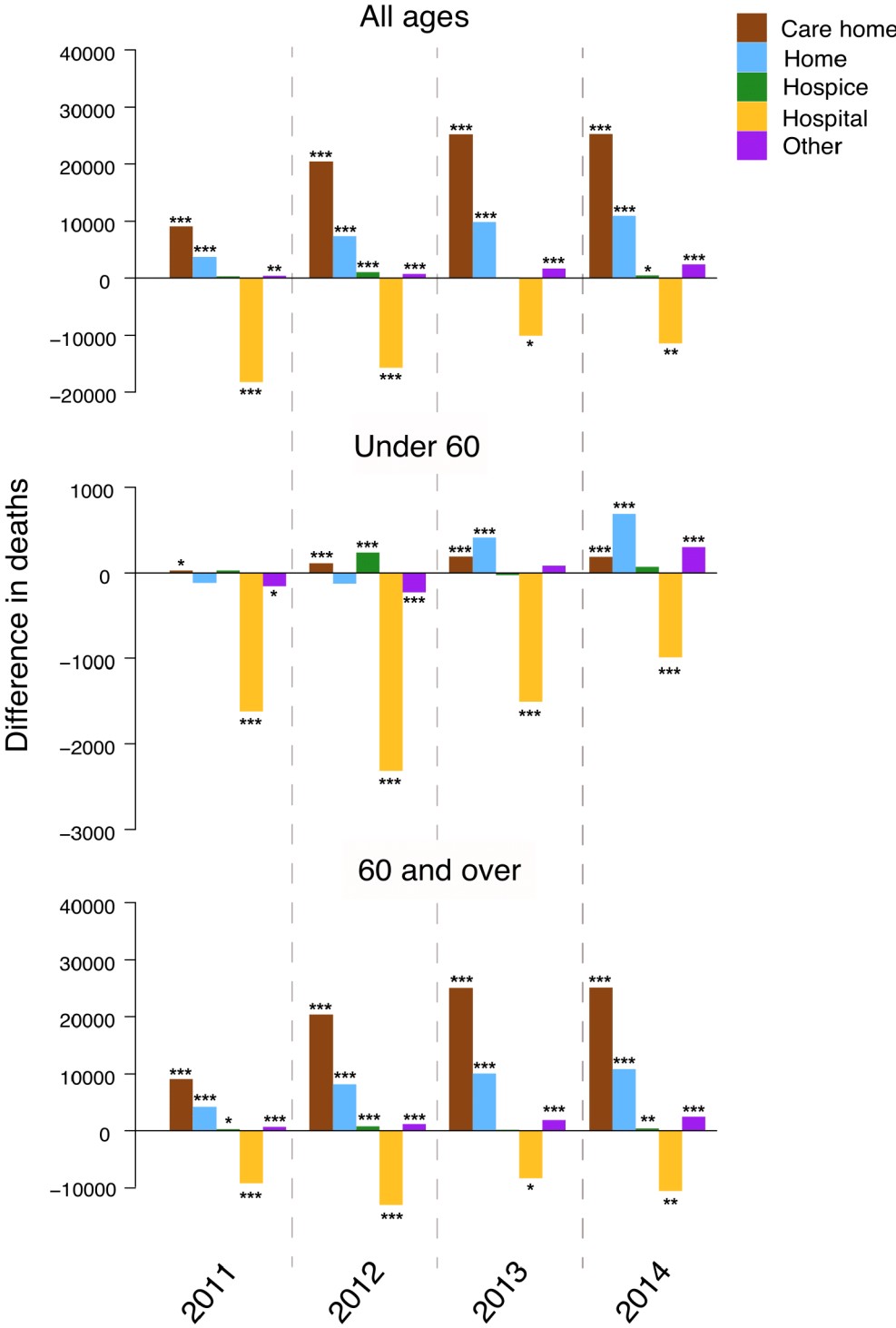

**Figure 2** Numbers of excess or lower than expected deaths for each place of death. Separate time trend analyses comparing actual to predicted mortality from 2011 to 2014 were conducted using mortality data categorised by place of death. Contributions from each place of death are colour coded. Data are shown for mortality rates for all ages (top panel), those under 60 (middle panel) and those 60 years or over (bottom panel). *p<0.05; **p<0.01; ***p<0.001.

used a regression model with both real PES and PEH per capita as explanatory variables and found only PES remained inversely associated with care home mortality (table 1). In this model, the variance inflation factor between PEH and PES was 4.15, suggesting that multicollinearity was not a problem. The results for home deaths replicated those for care home deaths (online

supplementary table S6). In contrast to care home and home mortality, real PEH per capita was more strongly related to hospital mortality than real PES per capita without adjusting for controls. On adjusting for macroeconomic forces, the relationship between PEH and hospital mortality was no longer apparent (online supplementary table S7).

**Table 1** Associations between public expenditure on health (PEH) or social care (PES) and care home deaths. Analyses were performed for 0–2 years of interval between PEH or PES and subsequent care home deaths

| | Care home deaths per 100 000 persons | | | |
| --- | --- | --- | --- | --- |
| | PEH per capita (£10) | | PES per capita (£10) | |
| Lag* (year) | β (95% CI) | p Value | β (95% CI) | p Value |
| Model 1 | | | | |
| 0 | −0.19 (−0.38 to −0.01) | 0.05 | −5.10 (−6.54 to −3.65) | <0.0001 |
| 1 | −0.09 (−0.28 to 0.11) | 0.39 | −3.54 (−5.43 to −1.65) | 0.001 |
| 2 | 0.06 (−0.13 to 0.25) | 0.52 | −1.38 (−3.54 to 0.79) | 0.23 |
| Model 2 | | | | |
| 0 | −0.63 (−0.78 to −0.47) | <0.0001 | −5.21 (−6.49 to −3.93) | <0.0001 |
| 1 | −0.49 (−0.65 to −0.33) | <0.0001 | −4.34 (−5.70 to −2.99) | <0.0001 |
| 2 | −0.22 (−0.44 to −0.01) | 0.05 | −2.73 (−4.40 to −1.06) | 0.005 |
| Model 3 | | | | |
| 0 | −0.88 (−1.11 to −0.65) | <0.0001 | −5.59 (−7.06 to −4.11) | <0.0001 |
| 1 | −0.82 (−1.01 to −0.63) | <0.0001 | −5.24 (−6.60 to −2.26) | <0.0001 |
| 2 | −0.55 (−0.78 to −0.32) | 0.0002 | −3.82 (−5.39 to −2.26) | <0.0001 |
| Model 4 | | | | |
| 0 | 0.14 (−0.01 to 0.29) | 0.08 | −6.23 (−8.10 to −4.39) | <0.0001 |
| 1 | 0.37 (0.16 to 0.56) | 0.002 | −7.05 (−9.55 to −4.56) | <0.0001 |
| 2 | 0.60 (0.35 to 0.85) | 0.0001 | −7.40 (−10.31 to −4.49) | <0.0001 |

Number of observations (sex-years) per analysis is 28.
Model 1: unadjusted model.
Model 2: adjusted for basic state pension per week.
Model 3: adjusted for unemployment rate and consumer price index.
Model 4: PEH and PES were included in the same model.
*Lag year of which PEH or PES preceded mortality rates.

## Identification of resources potentially mediating the spending–mortality relationship

We found that the numbers of NHS hospital and community nurses, and NHS health and social care clinical support staff were each associated with care home mortality. These factors alleviated the relationship between real PES per capita and care home mortality at lag years 1 and 2, suggesting these staff numbers were important mediators of the spending–mortality relationship (table 2). We found the number of nurses, but not clinical support staff, was suggested to be a mediator of the relationship between spending (both PEH and PES) and home mortality rates for lag years 1 and 2 (online supplementary table S8). We also found the number of nurses to be the sole mediator of the relationships between PEH per capita and care home mortality (online supplementary table S9) and PEH per capita and home mortality (online supplementary table S10) for all years. From 2001 to 2010, the average annual increase in nurse numbers was 1.61% whereas from 2010 to 2014, the average annual increase was over 20-fold lower at 0.07% (online supplementary table S11).

## Mortality projections to 2020 and scenario modelling

We projected mortality rates to 2020 using two observation bases: one using 2001–2010 data and a second using 2009–2014 (2009 and 2010 were included to ensure a sufficiently large observation base). We found that the 2009–2014-based projection entailed an additional 152 141 deaths (aggregate 95% CI 134 597 to 169 685) from 2015 to 2020 compared with the 2001–2010-based projection (online supplementary figure S3).

To determine what it would take to close this gap above planned health and social care spending to 2020/2021, we modelled three different scenarios (figure 3 and online supplementary table S12). On top of the combined health and social care budget, as of the end of 2016, the aggregate spending and efficiency combinations required to completely close the mortality gap would be: an additional £29.56 billion (£25.74–£33.37 billion) for a conservative 0% annual efficiency increase; an additional £27.26 billion (£23.61–£30.91 billion) for a moderate 1% annual efficiency increase; and an additional £23.03 billion (£19.67–£26.39 billion) for an aggressive 3% annual efficiency increase (figure 3 with an annual breakdown in online supplementary table S12). Under an ideal scenario in which no additional spending would be needed to close the mortality gap, the annual efficiency increase to 2020 would need to reach 10.79% (7.8–14.46). In these scenarios, efficiency gains reflect improvements across the system required to meet

**Table 2** Associations of public expenditure on social care (PES) and resources as potential mediating factors with care home deaths. Analyses were conducted for 0–2 years of interval between PES and subsequent care home deaths

| Potential mediators | Lag* (year) | Care home deaths per 100 000 persons | | | |
| | | PES in £10 million | | Potential mediator (thousands)† | |
| | | β (95% CI) | p Value | β (95% CI) | p Value |
|---|---|---|---|---|---|
| No. of hospital doctors | 0 | −6.01 (−7.64 to −4.38) | <0.0001 | 0.21 (0.006 to 0.41) | 0.06 |
| | 1 | −6.66 (−8.65 to −4.38) | <0.0001 | 0.57 (0.32 to 0.83) | 0.0002 |
| | 2 | −6.85 (−9.03 to −4.67) | <0.0001 | 0.94 (0.64 to 1.24) | <0.0001 |
| No. of GPs | 0 | −6.01 (−7.66 to −4.36) | <0.0001 | 0.87 (−0.009 to 1.76) | 0.06 |
| | 1 | −6.52 (−8.53 to −4.51) | <0.0001 | 2.35 (1.25 to 3.45) | 0.0004 |
| | 2 | −6.28 (−8.42 to −4.15) | <0.0001 | 3.67 (2.43 to 4.90) | <0.0001 |
| No. of nurses | 0 | −4.49 (−6.38 to −2.59) | 0.0001 | −0.09 (−0.28 to 0.09) | 0.34 |
| | 1 | −0.84 (−2.69 to 1.01) | 0.39 | −0.40 (−0.58 to −0.29) | 0.0001 |
| | 2 | 1.80 (−0.12 to 3.73) | 0.08 | −0.48 (−0.67 to −0.29) | <0.0001 |
| No. of scientific, therapeutic and technical staff | 0 | −6.15 (−7.83 to −4.48) | <0.0001 | −0.20 (0.01 to 0.38) | 0.04 |
| | 1 | −6.89 (−8.96 to −4.82) | <0.0001 | 0.52 (0.28 to 0.76) | 0.0002 |
| | 2 | −7.12 (−9.31 to −4.93) | <0.0001 | 0.85 (0.59 to 1.11) | <0.0001 |
| No. of ambulance staff | 0 | −6.09 (−7.73 to −4.46) | <0.0001 | 2.27 (0.17 to 4.37) | 0.04 |
| | 1 | −5.94 (−8.11 to −3.77) | <0.0001 | 4.60 (1.78 to 7.42) | 0.004 |
| | 2 | −5.09 (−7.70 to −2.47) | 0.001 | 6.72 (3.14 to 10.29) | 0.001 |
| No. of clinical support staff | 0 | −5.01 (−6.74 to −3.38) | <0.0001 | −0.02 (−0.26 to 0.22) | 0.85 |
| | 1 | −1.72 (−3.75 to 0.31) | 0.11 | −0.42 (−0.69 to −0.14) | 0.007 |
| | 2 | −0.49 (−1.87 to 2.86) | 0.69 | −0.43 (−0.75 to −0.11) | 0.02 |
| No. of infrastructure support staff | 0 | −5.78 (−8.63 to −2.91) | 0.0006 | 0.09 (-0.23 to 0.40) | 0.59 |
| | 1 | −5.31 (−9.43 to −1.20) | 0.02 | 0.21 (−0.23 to 0.66) | 0.35 |
| | 2 | −6.55 (−10.71 to −2.40) | 0.006 | 0.62 (0.18 to 1.06) | 0.01 |
| No. of overnight beds | 0 | −5.56 (−7.10 to −4.19) | <0.0001 | −0.13 (−0.25 to −0.008) | <0.0001 |
| | 1 | −5.42 (−6.95 to −3.90) | <0.0001 | −0.34 (−0.48 to −0.21) | <0.0001 |
| | 2 | −4.38 (−5.92 to −2.84) | <0.0001 | −0.50 (−0.65 to −0.35) | <0.0001 |
| No. of social care staff with accommodation | 0 | −5.20 (−6.64 to −3.77) | 0.01 | 0.09 (−0.05 to 0.23) | 0.21 |
| | 1 | −4.12 (−5.94 to −2.29) | 0.01 | 0.22 (0.02 to 0.42) | 0.04 |
| | 2 | −2.03 (−4.21 to 0.16) | 0.01 | 0.22 (−0.03 to 0.47) | 0.09 |
| No. of social care staff without accommodation | 0 | −5.69 (−7.19 to −4.20) | <0.0001 | 0.02 (−0.0002 to 0.04) | 0.06 |
| | 1 | −5.52 (−7.21 to −3.83) | <0.0001 | 0.05 (0.03 to 0.07) | 0.0003 |
| | 2 | −4.67 (−6.30 to −3.09) | 0.009 | 0.08 (0.05 to 0.10) | <0.0001 |

*Lag year of which PEH or PES preceded mortality rates.
†Estimates are shown for each corresponding factor in the left hand column.
GP, general practitioner; PEH, public expenditure on healthcare; PES, public expenditure on social care.

respective spending constraints, while avoiding excess mortality.

## DISCUSSION

This study demonstrates that recent constraints in PEH and PES spending in England were associated with nearly 45 000 higher than expected numbers of deaths between 2012 and 2014. If these trends continue, even when considering the increased planned funding as of 2016, we estimate approximately 150 000 additional deaths may arise between 2015 and 2020. Combining these projected excess deaths and the observed deaths prior to 2015 translates to around 120 000 excess deaths from 2010 to 2017. Contemporaneous reductions in life expectancy and excesses in measures of preventable death both validated our mortality findings.

The excess deaths observed in our study corroborate recent evidence highlighting the reversal of declining mortality trends observed in England and Wales in the past decade.[13] Similar to findings reported by Hiam

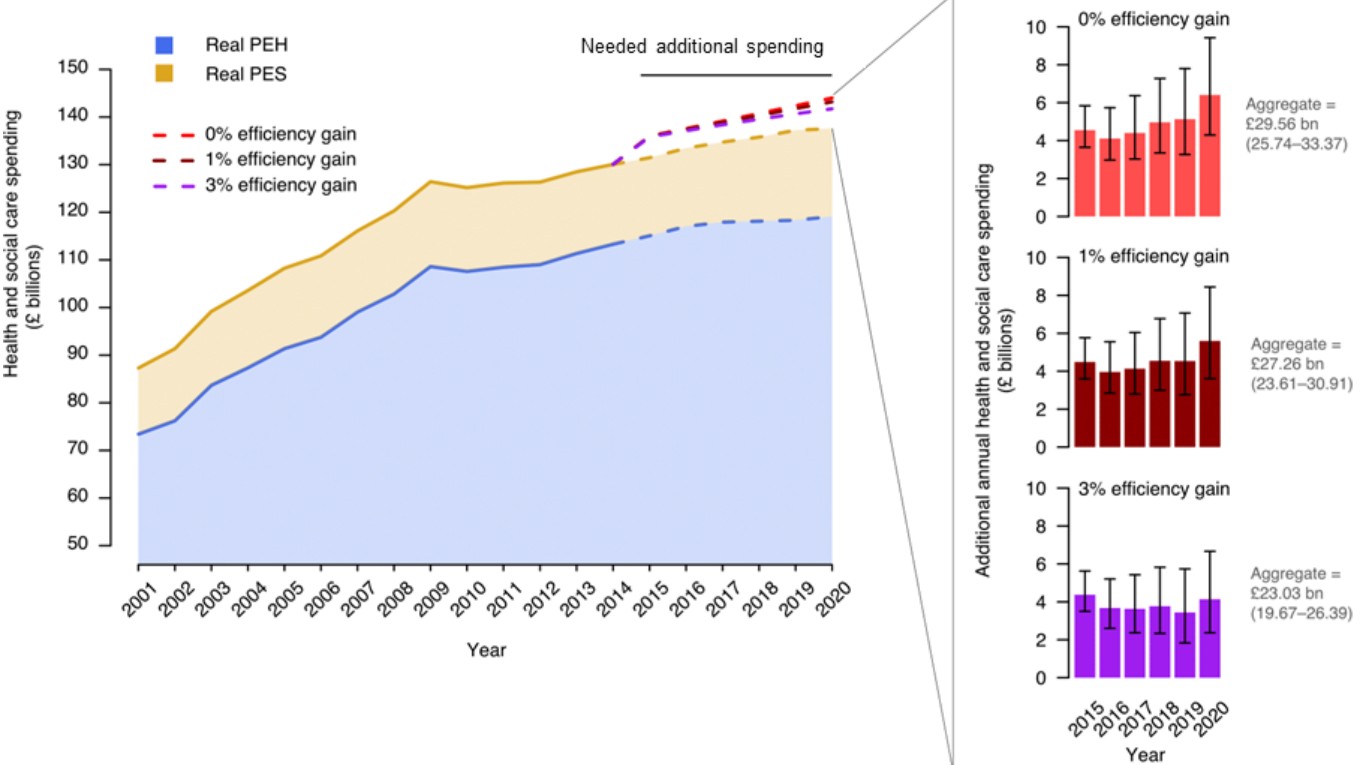

**Figure 3** Additional spending needed to close the 2020 mortality gap. Data are shown for PEH (blue) and on top of that, PES (beige), in real terms according to 2014/2015 prices. Actual out-turn data are shown for 2001/2002 to 2014/2015. However, for 2015/2016 to 2020/2021, the budgeted total Department of Health expenditure limit is shown for PEH. For PES, data are based on the continuation of −2.25% annual percentage change in core PES in 2014/2015 supplemented by the potential revenue from the adult social care precept for council tax.[12] Additional spending needed to close the projected mortality gap for each year from 2015 to 2020 is shown as color-coded dotted lines for three different scenarios, each of which assumes different annual efficiency improvements. The additional annual spending numbers with associated 95% CI are shown as bar plots for each scenario. PEH, public expenditure on healthcare; PES, public expenditure on social care.

and colleagues,[13] we observed that those aged ≥60 years are most susceptible to the observed excess mortality burden, as compared with those aged <60 years. We added to this evidence by additionally evaluating PYLL and life expectancy as outcomes, projecting excess deaths to 2020, determining potential mediating factors which may explain the observed excess deaths, and testing different hypothetical scenarios of funding and efficiency improvement which may be able to close the mortality gap.

By setting, deaths at care homes and at home contributed most to the observed 'mortality gap', while hospital mortality was lower than expected. This is most likely because social care experienced greater relative spending constraints than healthcare. Furthermore, the recent drive to move patients with poor prognoses and who have reached their ceiling of care away from the hospital environment to care homes or their own homes may have contributed to this.[25] It is also important to consider the architectural differences between health and social care delivery in England. The NHS provides publicly delivered universal health coverage, with studies suggesting this confers a protective effect during episodes of substantial cost constraint.[26] In contrast, social care is means tested

and often privately delivered, without universal coverage; factors that may influence access and quality.

The associations observed between PEH, PES and mortality were independent of macroeconomic changes that often co-occur with periods of reduced public sector spending. However, adjustment for potential mediating factors, nursing numbers in particular, nullified the associations between PEH, PES and mortality. Our study suggests that the number of NHS-qualified nurses is the strongest tested mediator of the relationships between spending, and care home and home mortality; this is congruent with findings in other reports.[27]

This study has several policy implications. First, it demonstrates that decelerated increases in PEH and PES, in England, may have adversely affected population mortality as demand increased and healthcare costs rose above inflation. This demonstrates that while health system design and ambition, such as delivery of universal coverage, is important, it must be adequately financed to ensure design translates into health improvement. Second, the finding that the elderly population and those in care homes were the most vulnerable to recent financial challenges, makes a strong case for targeted interventions to ensure adequate management of these patient

groups.[4] This includes funding increases in social care, in addition to maintenance or rises in nursing numbers aligned with demand. Third, and perhaps most importantly, there remains a prospective cost to the current trajectory of system financing that entails a number of excess deaths. While it would be presumptive to class these deaths as avoidable (we have used the terms 'additional' or 'excess' to describe higher than expected numbers; however it is not possible to determine the extent to which these deaths may be entirely preventable), we have outlined several funding efficiency scenarios that attempt to demonstrate how such a gap could perhaps be closed. Given that the health system has historically achieved 1%–2% annual productivity improvements, and that current demand is unprecedented, it seems unlikely that greater annual improvements could be expected. After factoring in planned government spending, and the £2 billion funding increase to social care announced in the 2017 Spring Budget, our analyses based on 1% annual productivity increase suggest that a cumulative spending increase of approximately £25.3 billion would be required to close this gap across health and social care by 2020/2021, equating to around £6.3 billion annually. Future studies may be needed to provide in-depth investigations on how funding allocations across PEH and PES may influence projected mortality rates.

This is the first study to provide an in-depth analysis of the potential effects of constraints in PEH and PES, on mortality. In this regard, we did not intend to analyse general changes in government policy, priorities or the overall environment in which health and social spending operates. A limitation was that our study was observational and retrospective, thereby our findings likely capture association rather than causation. Future studies combining different countries or regions with and without spending constraints may allow shock-based causal strategies or natural experiments which mimic random assignment in clinical trials. We were unable to analyse specific causes of death as outcomes because there were differences in how causes of death in 2001–2010 and 2010 onwards were coded, resulting in a lack of comparability for causes of death such as circulatory disease.[28] Finally, between 2010 and 2012, nurse numbers dropped by approximately 6000, which, by our regression analyses, translates to approximately 10% of expected deaths for that period. Therefore, the changes in NHS-qualified nurse numbers must only be partly responsible for the putative relationship between spending and mortality. Other resources and complex, emergent behaviour, which do not depend solely on the component parts of the health and social care system but rather on their interactions with one another, might also explain some of the additional deaths observed.

## CONCLUSION

We have found that spending constraints since 2010, especially PES, may have produced a substantial mortality gap in England. Our analyses demonstrate that if demand-side solutions are infeasible, large improvements in efficiency or, more feasibly, spending above growth in demand (and not just general inflation) are required to close this gap. We suggest that spending should be targeted on improving care delivered in care homes and at home; and maintaining or increasing nurse numbers.

**Author affiliations**
[1]Institute for Mathematical and Molecular Biomedicine, King's College London, London, UK
[2]PILAR Research and Education, Cambridge, UK
[3]MRC Unit for Lifelong Health and Ageing, University College London, London, UK
[4]Medical Sciences Division, University of Oxford, Oxford, UK
[5]Oxford University Clinical Academic Graduate School, John Radcliffe Hospital, Oxford, UK
[6]London School of Hygiene and Tropical Medicine, London, UK
[7]University of the Philippines Manila, Manila, Philippines
[8]University of the Philippines Diliman, Quezon City, Philippines
[9]Department of Applied Health Research, University College London, London, UK
[10]Department of Sociology, University of Cambridge, Cambridge, UK

**Acknowledgements** The authors would like to thank Mr Andrew Pring of Public Health England for extracting data used in the place-of-death analyses and Ms Lauren Passby and Ms Peggy Fooks for help regarding the initial data collection effort.

**Contributors** JW and MM conceived and designed the study. JW, WW, CDZ, DM and MM obtained the data. JW, WW, GDCS, PGDR and VAM conducted data formatting. JW and WW carried out statistical analysis with input from LPK. All authors helped interpret the findings. JW, WW and MM wrote the first draft of the manuscript with input from DCM, CDZ, RR and LPK. All authors provided input to subsequent drafts. All authors had full access to all of the data in the study and can take responsibility for its integrity and the accuracy of the data analysis.

**Funding** No funding was received for this study. WW is employed under Medical Research Council grant MC_UU_12019/2 and MC_UU_12019/4.

**Competing interests** MM is a co-founder of Cera, a technology-enabled homecare provider.

**Provenance and peer review** Not commissioned; externally peer reviewed.

**Data sharing statement** All data are in the public domain.

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
