## [Reviewer comments · BMJ Open]

ARTICLE DETAILS

TITLE (PROVISIONAL)	Effects of health and social care spending constraints on mortality in England: A time trend analysis
AUTHORS	Watkins, Johnathan ;Wulaningsih, Wahyu; Da Zhou, Charlie; Marshall, Dominic; Syliahteng, Guia; Dela Rosa, Phyllis; Miguel, Viveka; Raine, Rosalind; King, Lawrence; Maruthappu, Mahiben

VERSION 1 – REVIEW

REVIEWER	Elizabeth Bradley Yale University, USA
REVIEW RETURNED	17-May-2017

GENERAL COMMENTS	Overall, I am interested in this paper and think it is important but revisions are critical for reader understanding and clarity. Abstract: 1. Explain unit of fixed effects in abstract (the local authority); explain N2. Conclusion sounds too causal (produced) Introduction 3. “Real-term” social care – what does that mean?4. Some literature on social determinants of health in intro would be helpful (See Bradley, Taylor Paradox of American Health Care; Marmot; McGinnis) Methods 5. Explain what lower-tier LA is and why that is good to use. Why just them?6. Did any of the Las have shocks that one could use to get some causal modeling?7. P. 6 Lines 41-53 about the mediation and modeling needs more defense. The model building technique is not clear and should be explicit about how models were chosen.8. The years and gaps are also unclear in the methods. P. 7 top para. Results 9. Make clearer what is being compared longitudinally. Is it 2001/2002 versus 2009/2010 or do you have annual data? And how do 2010/11 v 2014/15 compare? The first para of Results could have a table to clarify. This problem of how many observation years do
--

	you have persists throughout. 10. Assessment of the mortality gap “using data on both sexes” - what data? 11. Sex-stratified time-series analyses were not described in methods. 12. The results do not follow the methods. Please be clearer and more systematic in the methods and then follow the same organization flow through to the results. 13. Did you check for multicollinearity between PES and PEH – how are they correlated? 14. Work force and bed numbers appear in the results but their sources are not in the methods as I read them. Maybe I missed them. 15. What do you mean by efficiency gain in the results? P. 11 Discussion 16. Limitations should be more prominent and be address. The causality problem is significant. Say more and how is this addressed or how should we interpret findings given design? 17. Measurement issues seem to be limitations as well – is everything measured the same way in spending across Las? Hard to believe. Are clinical outcomes really well measured? 18. Para bottom P. 12 is nice. 19. What are “health and social care resources? Spending? Be clearer. 20. Did not follow projections at end of discussion and seems that endogeneity could hamper the validity of such projections. Please address. Tables 21. Annotate these better. Hard to follow if they were separated from the paper.
--	---

REVIEWER	William Brinson Weeks, MD, PhD, MBA The Dartmouth Institute for Health Policy and Clinical Practice Lebanon, NH USA
REVIEW RETURNED	18-Jun-2017

GENERAL COMMENTS	This is a very well written and interesting paper that examines the impact of slowdown in spending on healthcare and social programs in Britain on mortality and finds 1. slowdown has resulted in greater than expected mortality 2. this slowdown has had most pronounced impact on those aged 60 and older 3. the impact there was greatest in nursing homes and 4. changes in social spending resulted in greater mortality increases than did those in healthcare spending. I have very few suggestions. First, I'd suggest highlighting model 4 in table 1 a bit more. If I'm reading that correctly, the greatest return to capital investment when considering both health and social services together is to social services and that investment there would have the greatest impact on mortality reduction. While that might throw a monkey-wrench in the authors' modeling of different required amounts to fill budget deficits, given different efficiency ratios, I think it would be worthwhile to determine the impact on budget when the desire is to keep mortality stable, for investment in EITHER PEH or PES....in other words, using model 4, how much money would need to be invested
--

	in PES to maintain steady mortality numbers if no additional money was invested in PEH and the converse (which, I believe, would actually be negative). In the longitudinal associations section and the identification of resources section, there are a few too many commas. Very nice, relevant, and interesting paper
--	--

VERSION 1 – AUTHOR RESPONSE

Reviewer: 1

Reviewer Name: Elizabeth Bradley

Institution and Country: Yale University, USA

Please state any competing interests: None Declared

Please leave your comments for the authors below

BMJ Open review

Effect of health and social spending...

Overall, I am interested in this paper and think it is important but revisions are critical for reader understanding and clarity.

Abstract:

1. Explain unit of fixed effects in abstract (the local authority); explain N

Response: Thank you for the Reviewer's comments. We have now provided more explanation in the Abstract's Methods as follows, "Fixed-effects regression analyses using annual data on PES and PEH with mortality as the outcome, with further adjustments for macroeconomic forces and resources. Analyses were stratified by age group, place of death, and lower-tier local authority (N=325). We also projected mortality rates to 2020 based on recent trends."

2. Conclusion sounds too causal (produced)

Response: We have now changed this into, "Spending constraints, especially PES, are associated with a substantial mortality gap."

Introduction

3. "Real-term" social care – what does that mean?

Response: We have now provided more explanation as follows, "Real-term adult social care spending decreased by 1.19% annually between 2010 and 2014 after correcting for the effect of inflation."

4. Some literature on social determinants of health in intro would be helpful (See Bradley, Taylor Paradox of American Health Care; Marmot; McGinnis)

Response: Thank you. We have now added the above references to highlight the role of social determinants of health in the Introduction as follows, "Although the role of social determinants in

health is increasingly acknowledged,¹ there is underinvestment in social care in many high-income countries such as the United States.²

Methods

5. Explain what lower-tier LA is and why that is good to use. Why just them?

Response: The following sentence has now been added to the Methods, "...325 lower-tier local authorities (LAs), which are the equal of districts, borough or city council..." We used lower-tier LAs as they allow a more granular view into potential variation on public spending and outcomes across England.

6. Did any of the LAs have shocks that one could use to get some causal modeling?

Response: Since spending constraints occurred nationwide, we were unable to identify any LA-specific shocks from our data sources which could support our analysis of the associations between spending constraints and mortality. We have acknowledged this in the Discussion as follows, "Future studies combining different countries or regions with and without spending constraints may allow shock-based causal strategies or natural experiments which mimic the random assignment in clinical trials."

7. P. 6 Lines 41-53 about the mediation and modeling needs more defense. The model building technique is not clear and should be explicit about how models were chosen.

Response: We have added the following sentences in the Methods to provide the rationale for the analysis, "Effects of public spending on resources such as staff or infrastructure have been documented, and these resources have also been linked with health outcomes. Therefore, we explored resources of health and social care as potential mediating factors using by running fixed-effects regression models with real PEH/PES per capita and each resource variable as the independent variables; and care home/home mortality as the dependent variable."

8. The years and gaps are also unclear in the methods. P. 7 top para.

Response: This has now been clarified as follows, "Two different mortality projection analyses for 2015 to 2020 were performed, each with a different observation base: one using 2001 to 2010 data and the other using 2009 to 2014 data, with the number of projected deaths based on 2001 to 2010 data subtracted from the number of projected deaths based 2009 to 2014 data."

Results

9. Make clearer what is being compared longitudinally. Is it 2001/2002 versus 2009/2010 or do you have annual data? And how do 2010/11 v 2014/15 compare? The first para of Results could have a table to clarify. This problem of how many observation years do you have persists throughout.

Response: Apologies for the confusion. We have now clarified that the population mortality data was collected annually as follows, "Annual population mortality data for England were extracted from the UK's Office for National Statistics (ONS)³ based on Medical Certificates of Cause of Death from the Registration Online system", whereas public spending data for a given year is based on financial year, which we have explained in an additional paragraph in the Methods starting as follows, "Nominal

public expenditure on health (PEH) data were defined as total expenditure limits for the Department of Health (responsible for the NHS in England), for the financial years 2001/2002 to 2014/2015..." To further the data being compared, and to explain how we addressed this limitation, we have added the following part in the Methods, "For a given year, population mortality on the same year was the outcome whereas spending data for a financial year starting at the given year was used as the main predictor. Since population mortality was collected annually and spending data is reported for each financial year (starting 1 April in the UK), we repeated our analysis using 1- and 2-year lag periods."

10. Assessment of the mortality gap "using data on both sexes" - what data?

Response: This has now been corrected into, "...we compared actual and predicted mortality rates for both sexes..."

11. Sex-stratified time-series analyses were not described in methods.

Response: We have now clarified this in the Methods as follows, "Analyses were stratified by sex and repeated for each age group, place of death, and local government area."

12. The results do not follow the methods. Please be clearer and more systematic in the methods and then follow the same organization flow through to the results.

Response: Thank you. We have now restructured the Methods and reorganise the Results section to follow the same organisation flow with the Methods.

13. Did you check for multicollinearity between PES and PEH – how are they correlated?

Response: We have now added this information in the Results as follows, "In this model, the variance inflation factor between PEH and PES was 4.15, suggesting that multicollinearity was not a problem."

14. Work force and bed numbers appear in the results but their sources are not in the methods as I read them. Maybe I missed them.

Response: Data sources for work force and bed numbers are available in the Supplementary Appendix as indicated in the Methods, "Further details, including health and social care resource data (staff and bed numbers), are provided in the Supplementary Appendix."

15. What do you mean by efficiency gain in the results? P. 11

Response: We have now replaced this with, "annual efficiency increase".

Discussion

16. Limitations should be more prominent and be address. The causality problem is significant. Say more and how is this addressed or how should we interpret findings given design?

Response: Thank you for mentioning this. We have now added the following sentence in the Discussion, "A limitation was that our study was observational and retrospective, thereby our findings reflect association rather than causation." We have further added a sentence detailing potential ways of addressing this limitation in future studies as mentioned above.

17. Measurement issues seem to be limitations as well – is everything measured the same way in spending across Las? Hard to believe. Are clinical outcomes really well measured?

Response: Thank you for mentioning this. There was discrepancy in the way data was reported, although this was evident for the consistency of measurements across the years rather than across LAs. For instance, causes of death in 2001-2010 and from 2010 onwards were coded with different software versions, which resulted in some discrepancy in specific mortality rates when used in the same data, e.g. a 5% decrease of deaths from the circulatory system between the old and new software versions.⁴ For this reason, we only reported findings on all-cause mortality instead of specific causes of death, and we have now added this limitation in the Discussion as follows, “We were unable to analyse specific causes of death as outcomes because there were differences in how causes of death in 2001-2010 and 2010 onwards were coded, resulting in a lack of comparability for a few causes of death such as circulatory disease”. There was also a change in the way resource data for social care was reported, and we have employed an established method to obtain comparable data between pre- and post-2009 assessments. We have mentioned this in the supplementary appendix as follows, “Prior to 2009, data was available from the Annual Business Enquiry, whereas the Business Register Employment Survey provided data from 2009 onwards under the SIC 2007 codes. Proportional mapping using ONS-published proportions between SIC 2003 and SIC 2007 was conducted to estimate the number of employment which corresponds to the aforementioned SIC 2003 codes between 2009 and 2014.⁵”

18. Para bottom P. 12 is nice.

Response: Thank you.

19. What are “health and social care resources? Spending? Be clearer.

Response: This sentence has now been changed into, “However, adjustment for potential mediating factors, nursing numbers in particular, nullified the associations between PEH, PES, and mortality.”

20. Did not follow projections at end of discussion and seems that endogeneity could hamper the validity of such projections. Please address.

Response: We have now deleted this part to avoid confusion.

Tables

21. Annotate these better. Hard to follow if they were separated from the paper.

Response: We have now revised our tables to provide a clearer description if separated from the paper.

Reviewer: 2

Reviewer Name: William Brinson Weeks, MD, PhD, M
BA

Institution and Country: The Dartmouth Institute for Health Policy and Clinical Practice, Lebanon, NH
USA

Please state any competing interests: None declared

This is a very well written and interesting paper that examines the impact of slowdown in spending on healthcare and social programs in Britain on mortality and finds 1. slowdown has resulted in greater than expected mortality 2. this slowdown has had most pronounced impact on those aged 60 and older 3. the impact there was greatest in nursing homes and 4. changes in social spending resulted in greater mortality increases than did those in healthcare spending.

I have very few suggestions.

First, I'd suggest highlighting model 4 in table 1 a bit more. If I'm reading that correctly, the greatest return to capital investment when considering both health and social services together is to social services and that investment there would have the greatest impact on mortality reduction. While that might throw a monkey-wrench in the authors' modeling of different required amounts to fill budget deficits, given different efficiency ratios, I think it would be worthwhile to determine the impact on budget when the desire is to keep mortality stable, for investment in EITHER PEH or PES....in other words, using model 4, how much money would need to be invested in PES to maintain steady mortality numbers if no additional money was invested in PEH and the converse (which, I believe, would actually be negative).

Response: We appreciate the Reviewer's valuable comments. We modelled different efficiency scenarios as a secondary analysis to provide an overview of how projected mortality rates may vary given different scenarios of productivity. Projecting the proportion of investment across PEH and PES to maintain current rates of mortality, albeit interesting, is beyond the scope of our study. We have now added this in our limitations as follows, "Future studies may also be needed to provide in-depth investigations on how funding allocations across PEH and PES may influence projected mortality rates."

In the longitudinal associations section and the identification of resources section, there are a few too many commas.

Response: We have now removed unnecessary commas and truncated long sentences in this section.

Very nice, relevant, and interesting paper

Response: Thank you

References

1. Marmot M. Social determinants of health inequalities. *Lancet*. 2005;365(9464):1099–1104.
2. Bradley EH, Sipsma H, Taylor LA. American health care paradox - high spending on health care and poor health. *Qjm* [Internet]. 2016;(October 2016):hcw187. Available from: <https://academic.oup.com/qjmed/article-lookup/doi/10.1093/qjmed/hcw187>
3. Office for National Statistics. Mortality statistics: deaths registered in England and Wales [Internet]. 2014. Available from:

<http://webarchive.nationalarchives.gov.uk/20160105160709/http://www.ons.gov.uk/ons/rel/vsob1/mortality-statistics--deaths-registered-in-england-and-wales--series-dr-/index.html>

4. Office for National Statistics. Statistical Bulletin Results from the ICD – 10 v2010 bridge coding study. London; 2011.

5. Office for National Statistics. Correlation between UK SIC 2003 and UK SIC 2007 [Internet]. 2015. Available from:

<http://webarchive.nationalarchives.gov.uk/20160105160709/http://www.ons.gov.uk/ons/guide-method/classifications/archived-standard-classifications/uk-standard-industrial-classification-1992--sic92-/index.html>

VERSION 2 – REVIEW

REVIEWER	EH Bradley Vassar College
REVIEW RETURNED	17-Aug-2017

The reviewer completed the checklist but made no further comments.